# Interobserver Variability Prediction of Primary Gross Tumor in a Patient with Non-Small Cell Lung Cancer

**DOI:** 10.3390/cancers14235893

**Published:** 2022-11-29

**Authors:** Wonjoong Cheon, Seonghoon Jeong, Jong Hwi Jeong, Young Kyung Lim, Dongho Shin, Se Byeong Lee, Doo Yeul Lee, Sung Uk Lee, Yang Gun Suh, Sung Ho Moon, Tae Hyun Kim, Haksoo Kim

**Affiliations:** Department of Radiation Oncology, Proton Therapy Center, National Cancer Center, Goyang 10408, Republic of Korea

**Keywords:** interobserver variability, deep learning, convolutional neural network, fuzzy set theory

## Abstract

**Simple Summary:**

To reduce interobserver variability (IOV) for primary gross tumor volume in a patient with non-small cell lung cancer (NSLCL), the concept of an IOV map was newly proposed using signed Euclidean distance transform, fuzzy set theory, and the IOV prediction network, which could predict an IOV map from the corresponding CT images. The clinical feasibility of reducing IOV with the predicted IOV map was evaluated using a two-dimensional Dice similarity coefficient, the Jaccard index, and Hausdorff distance. Our proposed method can reduce the IOV in a set of NSCLC patients and was statistically verified using a Wilcoxon signed rank test (*p* < 0.05).

**Abstract:**

This research addresses the problem of interobserver variability (IOV), in which different oncologists manually delineate varying primary gross tumor volume (pGTV) contours, adding risk to targeted radiation treatments. Thus, a method of IOV reduction is urgently needed. Hypothesizing that the radiation oncologist’s IOV may shrink with the aid of IOV maps, we propose IOV prediction network (IOV-Net), a deep-learning model that uses the fuzzy membership function to produce high-quality maps based on computed tomography (CT) images. To test the prediction accuracy, a ground-truth pGTV IOV map was created using the manual contour delineations of radiation therapy structures provided by five expert oncologists. Then, we tasked IOV-Net with producing a map of its own. The mean squared error (prediction vs. ground truth) and its standard deviation were 0.0038 and 0.0005, respectively. To test the clinical feasibility of our method, CT images were divided into two groups, and oncologists from our institution created manual contours with and without IOV map guidance. The Dice similarity coefficient and Jaccard index increased by ~6 and 7%, respectively, and the Hausdorff distance decreased by 2.5 mm, indicating a statistically significant IOV reduction (*p* < 0.05). Hence, IOV-net and its resultant IOV maps have the potential to improve radiation therapy efficacy worldwide.

## 1. Introduction

According to reports from the Health Insurance Review and Assessment Service and the 2020 Central Cancer Registry, lung cancer is the leading cause of cancer deaths in the Republic of Korea [1]. The number of patients treated for lung cancer in 2018 was 19,524, among which non-small cell lung cancer (NSCLC) accounted for 83.3% and the small cell variety accounted for 16.0%. Among the viable treatment methods, surgery was the most common (53.0%), followed by chemotherapy (31.6%) and radiation therapy (15.4%). Notably, radiation therapy still plays a major role in the treatment of inoperable NSCLC.

Owing to advances in radiation delivery techniques and medical imaging technology, the accuracy of radiation dose delivery with highly conformal delivery techniques, such as intensity-modulated radiation therapy (IMRT), volumetric modulated arc therapy (VMAT), and image-guided radiation therapy (IGRT), is approaching submillimeter levels of accuracy [2,3,4]. Notably, high-energy particle therapy can now achieve relatively high dose conformity with low collateral doses on organs at risk (OARs), owing to its Bragg peak characteristics [5,6,7]. Examples include intensity-modulated particle therapy (IMPT) [8,9,10] and ultrahigh-dose-rate (FLASH) radiotherapy [11,12,13]. Therefore, the importance of the effects of interobserver variability (IOV) of the gross tumor volume (GTV) on target volume coverage has recently been emphasized [14], as it can lead to an undesirable dose in OARs, which may lead to tumor recurrence [15,16].

To reduce IOV in volume delineation, various methods have been used to demonstrate that (1) group intervention with document guidelines and teaching, (2) provision of auto contour, and (3) acquisition of additional medical imaging are statistically effective in reducing IOV [17]. However, group intervention requires a time and place where participants can simultaneously gather me. If predicted auto contours are provided, IOV can be reduced. However, in addition to requiring a well-trained model, auto contour could provide limited information binarized into the foreground and background. In the case of the acquisition of additional medical imaging, effectively reducing IOV has been demonstrated using various techniques: computed tomography (CT), four-dimensional CT, cone-beam CT (CBCT), positron emission tomography (PET), and magnetic resonance imaging (MRI). However, acquiring additional medical images is labor-intensive and expensive. In addition, a standard protocol for image registration is required to extract clinically useful information. Moreover, accurate image registration techniques are often needed across multiple medical imaging devices.

IOV can be caused by various factors, such as differences in the application of oncological knowledge to individual patients, inadequate anatomical interpretation of cross-sectional anatomy on images, and variable knowledge of target volume definitions [17]. If multiple contours obtained from multiple radiation oncologists can be provided for each patient, it can help reduce IOV but is challenging in clinical settings.

We hypothesized that providing a two-dimensional (2D) map that quantifies and visualizes multiple contours reflecting the opinions of several radiation oncologists without additional medical imaging or labor would help reduce IOV.

To quantify IOV, fuzzy logic, which emerged in the context of fuzzy set theory, has been introduced to handle vagueness and uncertainty regarding various membership functions (e.g., triangular, trapezoidal, Gaussian, generalized bell, and sigmoid) [18].

Recently, deep learning models (i.e., convolutional neural networks (CNNs)) have been used to effectively learn the transformations between different domains, such as from CBCT to CT images [19,20], MRI to CT images [21,22], fluence to dose in water [23], and medical image to contour or mask [24,25,26,27,28,29,30], and their good performance has been validated by several studies. Furthermore, deep learning models show high performance in prediction and classification tasks performed by extracting high-dimensional features from various medical images [31,32].

In this study, we consider the likelihood of using IOV maps to reduce IOV. For this feasibility assessment, we focused on NSCLC patients’ primary gross tumor volume (pGTV) contours. First, to automatically produce the IOV maps, we propose an interobserver variability prediction network (IOV-Net), a novel CNN method based on fuzzy set theory that intakes CT images to predict the comparison IOV map. We leverage tumor contours manually delineated by five expert radiation oncologists to produce a ground-truth IOV map. Then, we compare them. Second, to test our method’s clinical feasibility for reducing overall IOV, we divide CT images among practicing oncologists at the National Cancer Center of the Republic of Korea (KNCC), and they create pGTV contours with and without the aid of the IOV map produced by the IOV-Net.

Both tests are highly successful, and we find that the IOV-Net and its resultant IOV maps have the potential to improve radiation therapy efficacy worldwide.

## 2. Methods and Materials

### 2.1. Multiple Delineation Database 

The multiple delineation dataset is publicly available at https://wiki.cancerimagingarchive.net/display/Public/NSCLC-Radiomics-Interobserver1. This dataset was constructed to analyze the effect of the PET-CT-based auto-segmentation of the GTV and involved the nodal volume on pathology and IOV [33,34]. The set includes CT images with radiation treatment (RT) structure data for 21 consecutive patients with histologically proven NSCLC. The clinical stage was Ib–IIIb. The RT-structure data included pGTV data that were manually delineated with and without the assistance of a source-to-background-based PET CT auto-segmentation tool by five radiation oncologists. These oncologists had different clinical experiences: three experienced thoracic clinicians and two residents. The CT images had 512 × 512 × (154–178) voxels with a voxel size of 0.98 × 0.98 × 5.0 mm^3^. The mean and median volumes of the GTV were 75.3 and 34.6 ccs, respectively. The range was 2.1–343.7 cc. In our study, CT images and manual delineation data were used without assistance (Figure 1).

### 2.2. IOV Map: Ground Truth

The ground-truth IOV map was created to provide information representing the IOV of NSCLC pGTV. The intensity of an IOV map ranged from 0.0 to 1.0; ours was closer to 1.0, and less IOV was estimated at the pixel of the CT slice. Moreover, if the intensity of the IOV map was closer to 0.0, the IOV was estimated to be greater.

The process of creating an IOV map was implemented slice-by-slice for each patient. Manual delineation data were converted into binary and reversed binary maps. Regarding the former, the pGTV region was assigned a value of one, and the others (background) were assigned values of zero. Otherwise, the values of the pGTV and the background on the reversed binary map were zero and one, respectively. Next, a 2D signed Euclidean distance transform was applied to the binary map and reversed binary map to create a distance map [35], which was summed into an accumulated distance map and reversed distance map, respectively (Equations (1) and (2)). Finally, an IOV map was created according to Equation (3). The fuzzy membership values were computed using a Gaussian membership function with a zero mean and 15.0 standard deviation. The parameters of the Gaussian membership function were empirically determined. Note that the Gaussian membership function differs from a Gaussian probability distribution in that the maximum value of the Gaussian membership function is 1.0, which is called a “fuzzy singleton.” The process of creating an IOV map is illustrated in Figure 2.
(1)$$AccDistanceMap=\;\sum_{n=1}^{5}signedEDT\left(BinaryMap_{n}\right)$$
(2)$$AccReversedDistanceMap=\;\sum_{n=1}^{5}signedEDT\left(ReversedBinaryMap_{n}\right)$$
(3)Interobserver variability map=gaussmf(AccDistanceMap+AccReversedDistanceMap−1)
where *n* is the number of radiation oncologists participating in the multiple delineation dataset, and *signedEDT* is the 2D signed Euclidean distance transform. *AccDistanceMap* and *AccReversedDistanceMap* are the accumulated binary and reversed distance maps of manual delineation data, respectively, and *gaussmf* is the Gaussian membership function.

### 2.3. IOV Prediction Model: IOV-Net

We developed an IOV-Net based on a CNN. The backbone architecture is a 2D U-Net with a pseudo-three-dimensional (3D) method. There are many variants of the U-Net. Vu et al. proposed a pseudo-3D method for medical image segmentation [28]. For the pseudo-3D method, the input includes multislice CT images, where adjacent slices are used as additional channels to the central slice. In the segmentation task, the output was a predicted single-slice binary mask corresponding to the central CT slice. The advantage of the pseudo-3D method is that it is much more computationally efficient than a fully 3D U-Net [29] and has better performance than a vanilla 2D U-Net [30].

In this study, five multiple CT slices (±two adjacent slices from the central slice) were used as input to the IOV-Net. The output is a predicted IOV map corresponding to the central slice. The IOV-Net has encoder and decoder structures. The encoder consists of a transition block, five convolution blocks, and four max-pooling layers. The transition block transforms the multislice input into a single-slice feature map using a 1 × 1 convolutional layer. The convolution block contains a 2D convolution layer, a 2D batch normalization (BN) layer, a rectified linear unit (ReLU), a 2D convolution layer, a 2D BN layer, and a ReLU sequentially.

The decoder consists of four upsampling layers, four convolution blocks, and a multiple-size convolutional block. The upsampling layer adopts the 2D nearest-neighbor method. The multiple-size convolution block is located at the last part of the IOV-Net, which consists of a 1 × 1 convolution layer that reduces the number of channels from 64 to 1, a 3 × 3 convolution layer, and a 5 × 5 convolution layer. The architecture of the IOV-Net is shown in Figure 3.

### 2.4. Data Preparation for Training and Clinical Feasibility Validation

For training the IOV-Net, an input consisting of five multiple slices was prepared. The CT image was not divided into patches, but an entire axial slice was used without resizing. For preprocessing, *z*-score normalization was applied to the CT images per patient [36,37]. The *z*-score normalization makes it easier and faster to find the global minimum or a good local minimum during model training. The ratio of inputs with and without the pGTV contour was determined as 1:1 to solve the imbalance issue. The mean number of CT slices that contain pGTV contour is approximately 9, and the range is from 4 to 19 slices. To overcome the limited dataset, on-the-fly (active) data augmentation was applied.

We designed a comparative study to evaluate the clinical feasibility of the IOV map for IOV reduction. The multiple delineation dataset was divided into training and testing sets, and the data from 16 patients were randomly assigned to a training set. The remaining five patients were assigned to a testing set (Group A). Similarly, the 16- and 5-patient data were randomly assigned to the training and testing sets (Group B). The test data for Groups A and B did not overlap.

The manual delineation of the test set was conducted for Group A by the four KNCC radiation oncologists. All observers were blinded to the contours delineated by the other observers. In Group B, the predicted IOV map was provided for manual delineation using a treatment planning system (Eclipse; Varian Medical Systems, Palo Alto, CA, USA). The setting parameters for the CT window width and level were not provided.

To cross-validate the clinical feasibility of the IOV map, manual delineation was repeated six months later [33]. However, in the cross-validation (CV), the IOV map was provided to Group A but not to Group B.

### 2.5. Training and Validation of the IOV-Net

To train the IOV-Net, its initial parameters were randomly initialized. The model was optimized to predict an IOV map corresponding to the central CT slice of the input. The smooth-L1 loss (Equation (4)) between the predicted and ground-truth IOV maps was used as an objective function for the model. Specifically, the smooth-L1 loss can be interpreted as a combination of the L1 and L2 losses [38,39].
(4)$$Smooth\;L1\;loss\left(x,\;y\right)=\frac{1}{n}\sum_{i}z_{i},z_{i}=\{\begin{array}{c} 0.5\frac{{\left(x_{i}-y_{i}\right)}^{2}}{\beta },\;\;if\;\left|x_{i}-y_{i}\right|<\beta \\ \left|x_{i}-y_{i}\right|-0.5*\beta ,\;\;otherwise \end{array}$$
where (*x*, *y*) are the ground truth and predicted IOV maps, respectively, and *n* is the number of samples. Hyperparameter beta (*β*) in Equation (4) is the value for applying additional weight to the loss. The value of *β* was empirically determined to be 0.5. An adaptive momentum estimation (Adam) optimizer was employed with a learning rate of 0.0001 and a weight decay of 0.0002. To compute the running average of the gradient, *β*1 was set to 0.9, and *β*2 was set to 0.999 [40].

The IOV-Net was trained for 5000 epochs. The training data were augmented by flipping along the horizontal axis; the rotation ranged between −5 and 5°, shear images ranged from −0.05 to 0.05, a zoom with a factor between 0.85 and 1.15 was used, and additional small elastic deformations were present, yielding approximately 320,000 slices for training.

The K-fold CV provided a good indication of how well the IOV-Net was universalized for unobserved data. We performed a patient-wise K-fold CV method (K = 5), and the training data of Groups A and B were divided into five disjointed subsets. The final model was assembled by averaging the outputs of the five models. No post-processing was performed.

The IOV-Net was implemented in Python 3.6 using PyTorch 1.10 [41] with CUDA 11.2. All networks were trained and tested on a graphics processing unit (GPU) workstation with an Intel Xeon processor (Intel, Santa Clara, CA) with 96 GB of RAM and an NVIDIA GeForce GV100 GPU with 32 GB of memory (NVIDIA, Santa Clara, CA, USA). The time required to predict an IOV map from the input of CT images was approximately 1.5 s with the GV100 GPU.

### 2.6. Evaluation Metrics for the IOV-Net and Clinical Validation

Several metrics were employed to evaluate the accuracy of the predicted IOV maps and analyze the effect of the IOV map on IOV reduction in a clinic. The mean squared error (MSE) between the predicted and ground-truth IOV maps was used as a metric to evaluate the accuracy of the predicted IOV map.

To analyze the clinical feasibility of the IOV map for IOV reduction, the variation among pGTVs was calculated. The 2D Dice similarity coefficient (DSC), Jaccard index (JI), and Hausdorff distance (HD) were employed to calculate the variation. However, a ground-truth contour of the pGTV was required to compute the three metrics. Thus, we estimated a true contour from the contours manually delineated by the four radiation oncologists using the Simultaneous Truth and Performance Level Estimation [42] algorithm included in the Computational Environment for Radiotherapy Research software [43]. The true contour was estimated using the expectation-maximization process with a 70% confidence level and was used as a reference map (pGTV-70) for comparison with each pGTV. Thus, the 2D DSC, JI, and HD were calculated from the reference map and pGTV contours.

For the statistical analysis, the results of the clinical evaluation were compared using a Wilcoxon signed rank test because our data do not satisfy the normality assumption according to the Shapiro–Wilk test [44]. All statistical analyses were performed using SAS software (v.9.4; SAS Institute Inc., Cary, NC, USA), and the statistical significance level was set at *p* = 0.05.

## 3. Results

### 3.1. Evaluation of the Prediction Accuracy of the IOV-Net

The IOV-Net was optimized to transform the input CT images into a predicted IOV map. To evaluate the accuracy of this task, the MSEs between the predicted and ground-truth maps were calculated for testing Groups A and B. According to the CVs, the MSEs of the five models were 0.0045, 0.0031, 0.0037, 0.0039, and 0.0041. The standard deviation of the mean MSEs was approximately 0.0005. In a repeated study, the MSEs of the five models were 0.0033, 0.047, 0.036, 0.032, and 0.0039. The standard deviation of the mean MSEs was again 0.0005. Among the test data, the samples of a central slice of the input, ground-truth and predicted IOV maps, and 2D difference maps are shown in Figure 4.

### 3.2. Analysis of the Effect of the IOV Map on the Reduction of IOV in Clinics

The pGTV in Group A was delineated without guidance. Otherwise, the pGTV of Group B was delineated using the guidance of the predicted IOV map from the initial study.

The DSC, JI, and HD between the four pGTVs and reference contours were calculated slice-by-slice. The means from Group A (with guidance) were 0.89, 0.84, and 6.65 mm, respectively. The means from Group B (without guidance) were 0.84, 0.79, and 6.64 mm, respectively.

In the repeated study after six months, manual delineation was performed for Groups A and B by four radiation oncologists. However, the IOV map was only provided to Group B. The means of the DSC, JI, and HD of Group A (without guidance) were 0.86, 0.80, and 8.26 mm, respectively. The values of Group B (with guidance) were 0.90, 0.84, and 6.15 mm, respectively. See Table 1 for the full listing.

As a statistical result of the integrated data of the two studies, the DSC (*p* < 0.0001), JI (*p* < 0.0001), and HD (*p* = 0.0356) were significantly different, depending on the presence or absence of the guidance of the predicted IOV map.

## 4. Discussion

With recent advancements in dose-delivery techniques, inconsistencies in the target pGTV contours delineated by observers can weaken the advantages of the new tools (e.g., IMRT, VMAT, IGRT, and IMPT). Regardless of the causes of these inconsistencies, there is an urgent need to reduce IOV.

This study was designed to evaluate the clinical feasibility of our proposed IOV-Net’s predicted IOV map as guidance for observer contour annotation. Four radiation oncologists working at KNCC manually performed NSCLC pGTV contouring with and without the guidance of the IOV map for a comparative study. We believe the fuzzy values on the IOV map could provide a guideline for the radiation oncologists during the pGTV contouring. A relatively low fuzzy value can make radiation oncologists more cautious, and a relatively high fuzzy value can increase the confidence of radiation oncologists. We found statistically significant differences in the resultant IOV depending on the presence or absence of the IOV map during pGTV contouring, apart from one of the four, who showed no statistical difference for DSC, JI, and HD (Table 2). The different trends among the four may be related to different clinical perspectives, anatomical knowledge, familiarity with patterns of pathological spread and recurrence, and basic experience [17].

The two main contributions of this study are as follows:IOV-Net, which applies an innovative implementation of the signed Euclidean distance transform and fuzzy membership function to predict the IOV map from input CT images without the need for additional human resources, image data, and time;A clinically effective method of reducing the IOV of clinician pGTV contour annotations, as validated for a set of NSCLC patients.

The limitations of this study stem from its use of published data and the size of the dataset. To overcome these limitations, we used an active data augmentation technique in the IOV-Net training procedure, and a K-fold (K = 5) CV technique was used to evaluate the generalized performance of the model. Additionally, to demonstrate clinical feasibility, the two studies were designed to have a time interval period of six months to take advantage of the human forgetting curve [45]. Nevertheless, IOV-Net was well-trained in a limited environment because it does not require individual pixel classifications for foreground vs. background, unlike mask and contour methods. Moreover, the CT images taken from the multiple delineation dataset were standardized in terms of anatomy and target location. Hence, the task required a relatively small number of data items. As a further study, the generalizability of the model to multiple types of tumors was investigated with more observers because one of the four seems to be an outlier.

This study aimed to clinically validate the efficacy of using an IOV map for guidance during practitioner contour annotation based on a map predicted by the IOV-Net deep learning model. For this assessment, NSCLC patient CT image data were used for training and validation. We believe that the proposed IOV-Net and the use of the resultant IOV map will contribute to reducing radiation toxicity and tumor recurrence while improving treatment outcomes by reducing IOV among oncologists. Future studies should consider collecting multi-observer manual delineation data so that the concepts of the IOV map may be applied to tumors at other sites or OARs.

## 5. Conclusions

We validated the clinical feasibility of the IOV map and the IOV-Net for reducing the IOV on the pGTV contours of a patient with NSCLC. The concept of an IOV map was newly defined in this study to quantify the IOV of contours, and the IOV-Net was successfully trained to predict an IOV map from a series of CT slices. Finally, the predicted IOV map was demonstrated to reduce the IOV on pGTV contouring in patients with NSCLC, and its clinical feasibility was statistically verified. Practically, we obtained an IOV map containing IOV information from CT without additional labor and time and showed that the IOV map can reduce IOV in the pGTV delineation process of lung patients. In the future, we plan to expand this work to consider other sites with a large number of radiation oncologists in a cohort study, including the breast, head, and neck, and integrate IOV-Net into the treatment planning system.

## Figures and Tables

**Figure 1 cancers-14-05893-f001:**
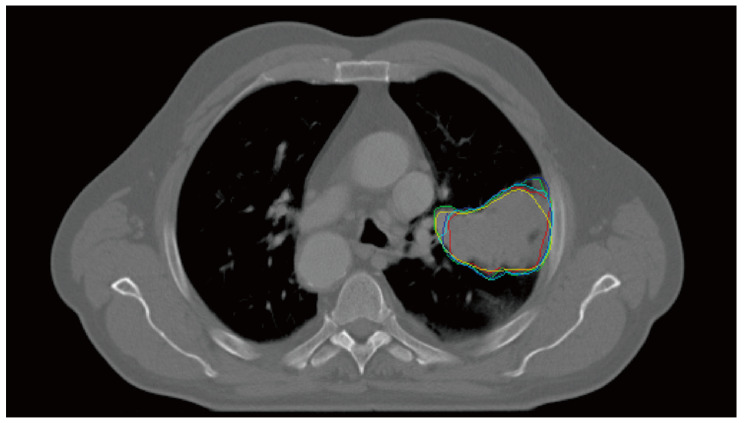
A computed tomography slice and the corresponding primary gross tumor volumes (colored solid lines) independently delineated by five radiation oncologists.

**Figure 2 cancers-14-05893-f002:**
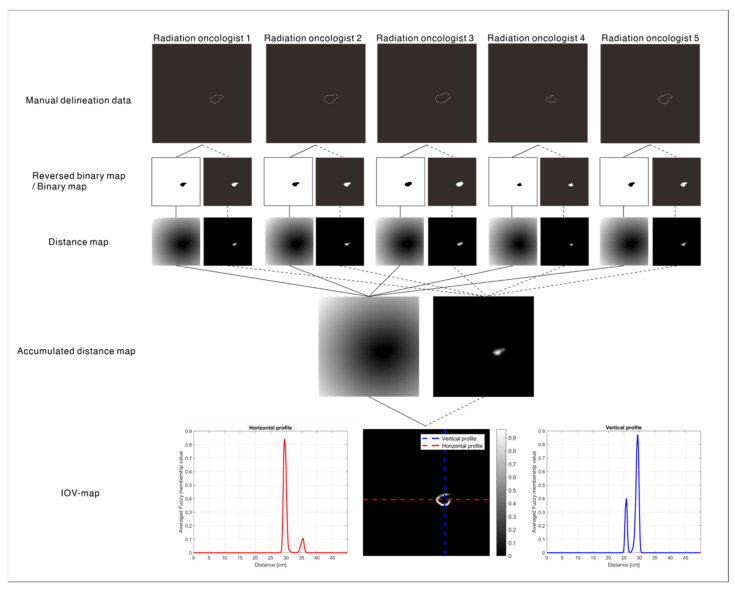
Procedures for creating an interobserver variability (IOV) map from five manual contours and the horizontal and vertical profiles of the IOV map.

**Figure 3 cancers-14-05893-f003:**
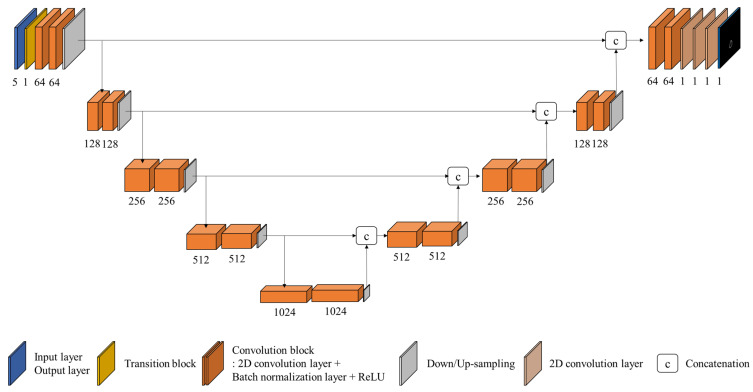
Diagram of the interobserver variability prediction network (IOV-Net) based on the U-Net with the pseudo-3D method.

**Figure 4 cancers-14-05893-f004:**
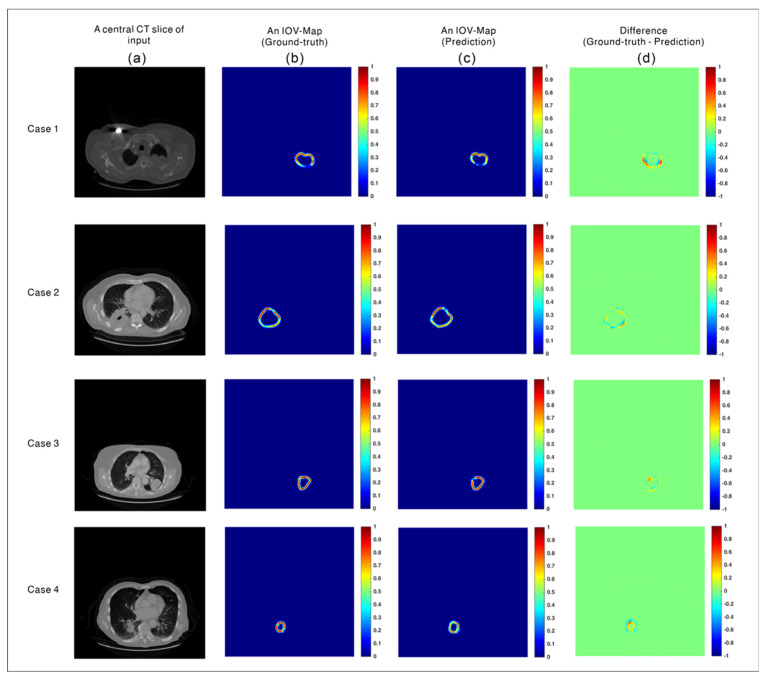
Samples of the test case for interobserver variability (IOV) map prediction: (**a**) central slice of the input, (**b**) ground-truth IOV map, (**c**) predicted IOV map, and (**d**) difference between the ground-truth and predicted IOV maps.

**Table 1 cancers-14-05893-t001:** Comparison results of the Dice similarity coefficient, Jaccard index, and Hausdorff distance of the initial and repeated studies for the test data.

**Initial Study**	**CT with the Predicted IOV Map ***			**CT Only**		
Radiation oncologists	DSC	JI	HD	DSC	JI	HD
A	**0.96**	**0.93**	**3.34**	0.91	0.87	4.68
B	**0.91**	**0.84**	**8.26**	0.82	0.76	7.51
C	**0.94**	**0.91**	**4.72**	0.78	0.73	7.32
D	0.76	0.68	10.60	**0.86**	**0.79**	**7.12**
**Repeated Study (Six Months Later)**	**CT Only**			**CT with Predicted IOV Map**		
Radiation oncologists	DSC	JI	HD	DSC	JI	HD
A	0.95	0.92	4.60	**0.97**	**0.95**	**3.74**
B	0.82	0.72	13.54	**0.85**	**0.77**	**7.24**
C	0.91	0.86	5.86	**0.94**	**0.89**	**5.95**
D	0.77	0.69	9.21	**0.85**	**0.76**	**7.74**

* Abbreviations: CT, Computed tomography; IOV map, Interobserver variability map; DSC, Dice similarity coefficient; JI, Jaccard index; HD, Hausdorff distance. The unit of the Hausdorff distance is mm. Bold values stand for the high scores in the two studies.

**Table 2 cancers-14-05893-t002:** Comparison results of the Dice similarity coefficient, Jaccard index, and Hausdorff distance for each radiation oncologist between the initial and repeated studies.

	DSC		JI		HD	
	Difference	*p*-Value	Difference	*p*-Value	Difference	*p*-Value
Radiation oncologist A	0.058 ± 0.217	**0.0220**	0.063 ± 0.229	**0.0287**	−1.631 ± 4.231	**0.0369**
Radiation oncologist B	0.051 ± 0.241	**<0.0001**	0.057 ± 0.237	**0.0002**	−2.819 ± 9.896	**0.0433**
Radiation oncologist C	0.094 ± 0.261	**<0.0001**	0.108 ± 0.248	**<0.0001**	−1.375 ± 5.542	**0.0217**
Radiation oncologist D	0.0 ± 0.196	0.5173	−0.004 ± 0.197	0.4687	0.649 ± 6.377	0.3260
Total	0.051 ± 0.232	**<0.0001**	0.056 ± 0.231	**<0.0001**	−1.042 ± 6.908	**0.0356**

Abbreviations: DSC, Dice similarity coefficient; JI, Jaccard index; HD, Hausdorff distance. The unit of the Hausdorff distance is mm. Bold values stand for the Wilcoxon signed rank test *p*-value < 0.05.

## Data Availability

The data presented in this study are available at https://wiki.cancerimagingarchive.net/display/Public/NSCLC-Radiomics-Interobserver1. For source codes of IOV-Net and IOV map, please contact the corresponding author.

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
