# Peer review of "Interobserver Variability Prediction of Primary Gross Tumor in a Patient with Non-Small Cell Lung Cancer"

_cancers, 2022, doi:10.3390/cancers14235893_

Round 1

Reviewer 1 Report

Authors introduce a new concept of IOV-map to reduce contouring interobserver variability of the primary gross tumor volume for non-small cell lung cancer patients. In the study, deep learning models are trained and tested to predict an IOV-map from the corresponding CT images. The authors evaluate the clinical feasibility of reducing interobserver variability when the contouring is performed with the guide of the predicted IOV-map

The manuscript is well written and figures well describe the study design. However, there is some points that need to be clarified or improved.

Materials and methods:

- It is not clear how the map was used to help the physician during contouring, the authors should better explain this aspect.

- Authors should also more fully justify the data augmentation process and specify why the chosen ratio of original to final dataset size will result in the best performing model.

- In order to  perform clinical validation, the authors assume a normal distribution of differences across the two related groups, a normality test should be carried out.

Finally, there are some typographical and formatting errors throughout the manuscript that suggest that the manuscript must to thoroughly proof-read. For example, in line 177 “Eq. 3” refers to eq. 4

Author Response

We are deeply grateful for your constructive comments, which have helped us enormously in improving the quality of the manuscript. We have carefully reviewed the comments of the referees and revised the manuscript accordingly. Our detailed response to the comments is given below.

Reviewer 2 Report

The paper is generally well-written, and the idea is effectively conveyed. This paper needs careful revision. The authors should revise the work by considering the reviewer's recommendations.

In the introduction, what key theoretical perspectives and empirical findings in the main literature have already informed the problem formulation? What major, unaddressed puzzle, controversy, or paradox does this research address?

Authors should further clarify and elaborate novelty in their contribution.

I fail to see any reference to the availability of the models, the data or the source code. Without this information it is impossible to independently verify or reproduce any of your claims, and the article greatly suffers in its utility and credibility.

A few abbreviations are not elucidated on the first appearance. Moreover, the initial letters for each acronym must be consistent (All capital is preferable, in my opinion).

Authors focus on article references, though they are many good recently published books and chapters that can be cited.

Below papers have some interesting implications that you could discuss in your introduction and how it relates to your work.

Vulli, A.; et al.. Fine-Tuned DenseNet-169 for Breast Cancer Metastasis Prediction Using FastAI and 1-Cycle Policy. Sensors 2022, 22, 2988.

El-Sappagh, Shaker, et al. "Automatic detection of Alzheimer’s disease progression: An efficient information fusion approach with heterogeneous ensemble classifiers." Neurocomputing (2022).

The conclusion section is concise. There is a need to work on it further.

Future work is better added to the conclusion section.

What are the limitations of the present work?

What are the practical implications of this research?

Author Response

(The authors gave the same response as above.)

Round 2

Reviewer 2 Report

.